# Fast and Efficient Simulation of the FEBID Process with Thermal Effects

**DOI:** 10.3390/nano13050858

**Published:** 2023-02-25

**Authors:** Alexander Kuprava, Michael Huth

**Affiliations:** Institute of Physics, Goethe University, 60438 Frankfurt am Main, Germany

**Keywords:** FEBID, nanofabrication, simulation, Monte Carlo, continuum model, electron beam

## Abstract

Focused electron-beam-induced deposition (FEBID) is a highly versatile direct-write approach with particular strengths in the 3D nanofabrication of functional materials. Despite its apparent similarity to other 3D printing approaches, non-local effects related to precursor depletion, electron scattering and sample heating during the 3D growth process complicate the shape-true transfer from a target 3D model to the actual deposit. Here, we describe an efficient and fast numerical approach to simulate the growth process, which allows for a systematic study of the influence of the most important growth parameters on the resulting shape of the 3D structures. The precursor parameter set derived in this work for the precursor Me_3_PtCpMe enables a detailed replication of the experimentally fabricated nanostructure, taking beam-induced heating into account. The modular character of the simulation approach allows for additional future performance increases using parallelization or drawing on the use of graphics cards. Ultimately, beam-control pattern generation for 3D FEBID will profit from being routinely combined with this fast simulation approach for optimized shape transfer.

## 1. Introduction

Over the last decade, focused electron- and ion-beam-induced deposition (FEBID and FIBID, respectively) have experienced significant advancements and have been shown to be among the most promising methods for three-dimensional (3D) free-form nanofabrication [1]. For FEBID, in particular, with controlled feature sizes in two-dimensional (2D) direct-write fabrication reaching below 10 nm [2,3] and its capability to reach feature sizes below about 50 nm in 3D structures with various materials [4,5,6] lets it stand out with regard to functional material selection and resolution. For example, structures fabricated by the FEBID technique have been shown to possess remarkable superconducting [7] and magnetic properties [8]. Several works have been published dedicated to studying the magnetic properties of these structures grown by FEBID, particularly those exhibiting new types of magnetic configurations induced by curved geometries [9,10,11,12,13,14]. In several of the latest studies, the recently developed precursor HFeCo_3_(CO)_12_ was utilized due to its ease of use, chemical stability and the associated improved control over the deposition process in comparison to such precursors as, e.g., Co_2_(CO)_8_ [15]. The resulting FeCo_3_ structures exhibit up to 80% at-% of metal content for 2D deposition [16] and are purely metallic when growing 3D structures [17].

With regard to 3D FEBID fabrication of the structures starting from a 3D CAD model, shape-true transfer is one of the main challenges [18]. Conventionally, FEBID is used for planar structure fabrication comprising, e.g., squares, circles or lines that serve as electrical contacts or patches. This can be relatively easily achieved by well-established rastering geometries that follow the desired shape. The deposition of such geometries can be simulated reliably by using a rather simple 2D hybrid Monte Carlo-continuum model simulation [19]. However, with the extension of the deposits into 3D and targeting more complex shapes, undesired surface artifacts occur [6,19], which are caused by temperature and proximity effects [6,19]. In the case of more complex structures, shape-true deposition of a desired 3D shape requires a careful selection of optimized patterning strategies that take into account several 3D growth implications [19,20]. In fact, without the assistance of a model-to-pattern software tool (in short, *slicer*), pattern development involves a laborious experimental process of adjusting the rastering pattern by trial and error. 

Several works have been published addressing this problem by introducing slicing and pattern optimizing algorithms able to analyze wireframe- and sheet-like structures and allow for compensation of diffusion, heating and proximity effects [4,21,22]. Recently, a Python tool was developed that allows the application of height correction, temperature compensation and proximity correction for sheet-like model structures [21]. Another tool that addresses wireframe-like structures was presented in [22] by J. Fowlkes. The user interface allows the construction of wireframe structures from which a pattern file can be obtained to be used by the SEM patterning software. Previously, a pattern file generator for wireframe-like structures was also developed by L. Keller that applies height and proximity corrections [4]. Recently, a work dedicated to nanowire bending correction by beam deceleration was introduced [23]. The approach takes into account beam heating effects that are the main cause of the downward bending of tilted segments. 

Despite the fact that significant advancements have been made regarding compensation algorithms, true-shape transfer of 3D model structures still represents a substantial challenge in FEBID. To a significant degree, this is because of the strongly varying values for key parameters relevant to the growth process if one compares different precursors. Among these key parameters is the coefficient for precursor surface diffusion and its temperature dependence, the temperature-dependent average precursor residence time and the energy-averaged electron-induced dissociation cross-section [24]. These parameters are hard to estimate and, in most cases, cannot be obtained from independent direct measurements [24]. So far, the most often employed method of parameters estimation is a coupled experiment-simulation approach based on the continuum theory for FEBID (see below) that has been utilized to determine parameters for a small number of Cu-, W- and Pt-based precursors [24,25,26,27]. 

The latest FEBID simulation presented by J. Fowlkes tackles the shape-prediction challenge and, assisted by a simple computer-aided design (CAD) program, generates the beam rastering pattern necessary for the fabrication of a desired nanostructure of the wireframe type [6]. The hybrid Monte Carlo-continuum simulation presented in that work combines the continuum model [28] with electron–matter interaction simulation to describe the electron-beam-driven deposition process. The simulation consists of two major parts: a FEBID continuum simulator and a Monte Carlo electron–matter interaction module. The Monte Carlo module is used to generate a database of a statistically significant number of electron trajectories in a semi-infinite domain of a predefined material. In the FEBID simulation, module-saved trajectories are then randomly drawn from the database and adapted to the current shape of the deposit. Based on the energy loss of scattered electrons, secondary electron emission is simulated. The outcoming surface secondary electron flux is then used to calculate the surface precursor coverage and the amount of deposited material.

Here, we present a hybrid Monte Carlo-continuum model simulation approach that adapts the general algorithm and the solutions as used in the work of J. Fowlkes and also takes into consideration earlier versions of the simulation approach [24,29,30,31]. We go beyond these previous works by explicitly taking temperature changes inside the growing 3D structure into account, which can become prominent if heat conduction in the structures is small, as was shown before [20]. The modular structure of the developed code allows for subsequent and incremental speed optimization employing code parallelization and transfer to GPUs. In addition, we demonstrate that the used effective model approach of the FEBID process has matured to a point of being a reliable foundation for simulations. The simulation approach used here can be foreseeably used for a wider range of tasks, including shape prediction, pattern-file development and optimization (i.e., simulation-assisted slicing), analysis of different patterning strategies, estimation of process parameters including precursor properties, simulating the shape geometry of hidden parts of hollow 3D models and more.

## 2. Materials and Methods

### 2.1. Simulation Overview

The fast FEBID simulation represents a direct-write material deposition simulation based on a chemical reaction (dissociation) driven by the electron irradiation. It consists of two major blocks working in turns: a Monte Carlo electron beam-matter module and a deposition process module (Figure 1). The simulation volume space discretization with equally sized grid cells is employed.

The Monte Carlo (MC) module handles complete electron–matter interaction by simulating the elastic scattering, continuous energy loss via inelastic scattering in between elastic scattering events and secondary electron emission. MC module yields a spatial surface electron (SE) flux distribution for the given shape of the solid. The module is executed first, and then every time, the beam moves to the next position or a complete cell is filled. The algorithm consists of three main subroutines. In the first subroutine, a series of electron trajectories are generated that undergo scattering in the given solid structure. The defined number of primary electrons (PE) sampled from a Gaussian beam profile undergo a series of scattering events until they reach a cut-off energy of 100 eV or escape the simulation volume, whereby their respective trajectories terminate. As electrons propagate through the solid, they continuously lose energy between each elastic scattering event. The sequence of lines between the elastic scattering events and energy loss on every segment composes an electron trajectory. In the second subroutine, the generated trajectories are superimposed on the grid, and the energy loss is accumulated inside the traversed cells. In the next subroutine, the energy loss is partially converted into SEs emitted along the PE trajectories. SEs that escape the solid are accumulated into the surface SE flux, which is used in the reaction-Equation (1) in the deposition process module. In the third subroutine, the MC module calculates the energy distributed inside the structure by primary electrons and scattered SEs that did not reach the surface. From the deposited energy, beam heating power, i.e., the associated temperature distribution within the solid is calculated.

In the deposition process module, the reaction equation, describing the precursor adsorption, desorption, diffusion and dissociation, is numerically solved to obtain the surface distribution of precursor from which the amount of additively deposited material due to precursor dissociation on the growing structure is calculated. Here, the surface SE flux generated in the MC module is used in the dissociation term of the reaction equation as well as for the calculation of the dissociated volume. These calculations are executed repeatedly during the dwell time, with a time step ∆t in order to simulate the time-dependent changes in precursor surface concentration and deposit evolution during the continuous exposure to the beam. Temperature changes are simulated via calculation of the 3D temperature distribution from beam-induced heating. The temperature distribution over the surface, in turn, influences surface diffusion and mean residence time of the precursor and, consequently, the precursor coverage.

Here, we refrain from going deeply into simulation code details and discuss only the general algorithms, key ideas and concepts used. A more detailed explanation can be found in the simulation documentation and code comments.

### 2.2. Deposition Module

The time-dependent surface precursor coverage is governed by the reaction–diffusion equation with the corresponding terms referring to precursor adsorption, desorption, dissociation and diffusion, respectively:(1)$$\frac{\partial n}{\partial t}=k\Phi \left(1-\frac{n}{n_{0}}\right)-\frac{n}{\tau }-\sigma fn+D\left(\frac{\partial^{2}n}{\partial x^{2}}+\frac{\partial^{2}n}{\partial y^{2}}+\frac{\partial^{2}n}{\partial z^{2}}\right)$$

The precursor surface flux *Φ* is given by the average diffuse component of the gas flux. Generally, it contains an additional directed component that is required to simulate shadowing effects caused by the orientation of the GIS with respect to the deposited structure [19]. In the simulation shown here, we refrain from taking the directed flux component into account, as the GIS positioning relative to the so-called hockey stick reference structure (see below) was reported to have negligible impact on the inclination angle [6]. 

The differential equation describing the precursor coverage is solved using the Runge–Kutta 4 method. The diffusion term is calculated by the explicit FTCS (Forward in Time Central in Space) scheme together with the Runge–Kutta method applied as well, which allows consideration of four nearest neighbors at a single iteration instead of just one. In order to ensure diffusive flow of the precursor is only on the surface, i.e., preventing diffusion into void or solid cells, a ghost cell method is utilized. The method is described in detail in Appendix A. In short, cells that neighbor a surface cell mimic precursor concentration value in that cell. It consequently sets concentration gradient between the cells to zero, nullifying diffusion flux.

The reaction Equation (1) is solved with a calculated time step that defines the iterative time progression. The main time step ∆*t* is set as the longest possible time step resulting in a stable numerical behavior and is derived from the shortest intrinsic time step of the three processes: *desorption*, *dissociation* and *diffusion*. Due to the possible variation in surface electron flux caused by the evolving topography in the BIR (beam interaction region) during deposition, the dissociation time step may vary. Diffusion and desorption rates increase with time as a result of beam heating, which requires a re-determination of the optimal time step. The mechanism of diffusion and desorption rate increase is described later.

During the simulation, the precursor surface coverage is calculated continuously and coupled with the deposited volume calculation. At a single time step, a cell is filled with a miniscule fraction of deposited material dependent on electron flux and local precursor coverage. In order to define the volume fraction ∆*V* deposited during a time step, it is necessary to know the volume of the non-volatile fragment of a dissociated precursor molecule. Considering, e.g., the PtC_5_ bulk composition of the deposit, this volume can be estimated based on the deposit density:(2)$$\Delta V=\frac{M}{N_{A}\rho }$$
where *M* is the molecular mass and *ρ* is the mass density.

### 2.3. Monte Carlo Module

#### 2.3.1. Single Scattering Model

The single scattering model was described by Joy [32]. This type of approach to the Monte Carlo simulation was chosen primarily due to its simplicity and accuracy at higher beam energies (>20 keV). We assume here a single-material deposition scenario in which each filled grid cell is associated with one specific deposit material, which influences scattering of the electrons in the solid medium. 

The model relies on two major assumptions: the elastic scattering of electrons determines their trajectories and thus the spatial scattered electron distribution, whereas the electrons lose energy continuously in-between elastic scattering events as a result of various inelastic scattering effects. The rate at which electrons lose their energy is calculated according to a modified Bethe expression: (3)$$\frac{dE}{dS}=-78,500\frac{\rho Z}{AE}\log\left(\frac{1.166\left(E+0.85J\right)}{J}\right)$$
where *ρ* is the material density, *Z* is the atomic number, *A* is the molecular weight, *E* is the electron energy and *J* is the ionization potential. 

The total elastic scattering cross-section is calculated using the modified Rutherford expression [33] that approximates the exact Mott values more accurately than the standard Rutherford expression:(4)$$\sigma^{T}=5.21\cdot 10^{-21}\frac{Z^{2}}{E^{2}}\;\frac{4\pi \lambda \left(1-e^{-\beta \sqrt{E}}\right)}{\alpha \left(1+\alpha \right)}{\left(\frac{E+511}{E+1022}\right)}^{2}$$
where *λ* and *β* are constant for a given element, *α* is the screening factor.

The number of electron trajectories simulated is set to 1000, which was previously shown to be statistically sufficient [6]. The outcoming data, namely the SE surface flux and distributed energy within the deposit volume, have both to be multiplied by a normalization factor:(5)$$f=\frac{i_{e}}{1000\cdot q_{e}}\;$$
where *i_e_* is the beam current and *q_e_* is the electron charge. 

For the purpose of tracking an electron trajectory and its intersections with the grid cells, a ray-tracing algorithm is utilized. It is described in detail in Appendix A.

#### 2.3.2. Parametric Secondary Electron Model

Each primary electron trajectory is subdivided into smaller segments to finely discretize the continuous energy loss. A fraction of the energy loss on each smaller segment is spent on the secondary electron emission according to the parametric formula:(6)$$n_{SE}=\frac{f\;dE}{\varepsilon \;ds}\Delta s,$$
where *n_SE_* is the number of emitted secondary electrons, ε is the energy required for the emission of a secondary electron, ∆*s* is the length of a trajectory subsegment. The values of *f* and ε were taken from [6]. The secondary electrons are emitted in a random direction so that *n_SE_* can be regarded as an isotropic SE source. By randomly picking the direction of a vector equal in length to the inelastic mean free path *λ* [34], SEs are emitted from the source position and then tested for traversing a surface voxel. If a vector traverses a surface voxel, the number of emitted electrons is associated with that voxel. The subsegment length or segment discretization ∆s is chosen such that a sufficient number of vectors pierce the surface to describe the spherically random direction distribution. The procedure results in a spatially resolved surface secondary electron emission profile *f* that is used in the reaction–diffusion Equation (1). In order to avoid unnecessary computations, segments at depths *z > 2·λ* below the surface are not considered during this procedure. They are bookmarked for the later calculation of beam heating (Section 2.4).

In our model, only secondary electrons contribute to the surface electron flux and thus influence the deposition process directly, while BSE, FSE and PE contribute to the dissociation process only indirectly through the secondary electrons that are emitted when PEs or BSEs pass through solid matter or the surface. 

As it was proposed previously [6], secondary electron refraction at the deposit–vacuum interface is ignored. Refraction influences the shape of the growing structure insignificantly and may only be needed to predict very fine sub-10 nm surface features.

### 2.4. Beam Heating

Thermally activated diffusion and desorption greatly influence surface precursor coverage and, consequently, the shape of the deposit. The growth of the deposit is therefore influenced by beam-induced heating and the associated local temperature increase. Temperature effects are more pronounced for materials with low thermal conductivity, such as unpurified deposits from the precursor MeCpPtMe_3_.

During exposure, the structure is subject to Joule heating through PE, FSE, BSE and SE. Firstly, PE, FSE and BSE lose energy continuously as they propagate through the solid medium according to the Equation (3). While the simulation distinguishes only primary and secondary electrons, the contributions of the three above-mentioned electron types are considered altogether. Only a fraction of the total energy loss is converted to heat:(7)$$Q_{PE}=E_{loss}\cdot \left(1-f_{SE}-f^{*}\right)=E_{loss}\cdot f$$
where *f_SE_* is the fraction of inelastic energy that is spent in SE creation and *f** is the fraction spent on other types of energy loss, such as X-ray emission. Assuming *f_SE_* = 0.6 and *f** = 0.01, *f* is equal to *0.4* in our simulation.

Secondly, SEs that do not escape the solid are inelastically scattered and their energy is also converted to heat. If an isotropic SE source *n_SE_* is buried deeper in the solid than the SE inelastic mean free path, it is considered scattered and contributes to heating. The dissipated energy yield is estimated based on the average energy of an SE in PtC_5_ deposits using *E_SE_ = 19* eV [6]:(8)$$Q_{SE}=n_{SE}\cdot E_{SE}$$

Then total heat generated by the beam is:(9)$$Q=Q_{PE}+Q_{SE}$$
where *Q, Q_PE_, Q_SE_* are spatially resolved and represent the distribution of volumetric heat sources inside the structure. The resulting heat profile is then used as power source term in heat Equation (10).

It has been previously shown that the only effective heat dissipation occurs through the substrate and heat dissipation through radiation (or convection) can be neglected for structures at the nanoscale level [20]. In our simulation, the substrate temperature is therefore fixed, i.e., the substrate serves as a heatsink with virtually infinite thermal conductivity and heat capacity. In the same work [20], it has been clearly shown that the heat diffusion process has a significantly shorter characteristic time scale than diffusion or adsorption/desorption. Thus, in contradistinction to precursor diffusion, the 3D temperature profile inside the growing structure reaches steady-state almost instantly after beam displacement and is governed by:(10)$$\nabla^{2}T=-\frac{Q}{k}$$

The derived state persists until a new cell is filled or the beam is displaced. Note that even after several consequently filled cells, the temperature increases just by a 1/100 of a degree due to the miniscule deposited volume for associated with the small cell sizes (2-10 nm). 

The resulting 3D temperature profile is updated at a rate that depends on the volumetric growth rate. Recalculation occurs at every 10,000 nm^3^ of the deposited volume, which, depending on the thickness, corresponds to 0.01–0.03 K resolution of the temperature profile evolution. The Simultaneous Over-Relaxation method (SOR) method is utilized for the temperature profile calculation. A more detailed explanation of the numerical solution can be found in Appendix A.

The dependence of residence time and diffusion coefficient on temperature follow an Arrhenius equation, respectively:(11)$$\tau \left(T\right)=\frac{1}{k_{0}}\cdot \exp\left(-\frac{E_{a}}{k_{B}T}\right)$$
(12)$$D\left(T\right)=D_{0}\exp\left(-\frac{E_{D}}{k_{B}T}\right)$$

The reaction–diffusion Equation (1) used to calculate surface precursor coverage becomes temperature-dependent through the two corresponding terms:(13)$$\frac{\partial n}{\partial t}=k\Phi \left(1-\frac{n}{n_{0}}\right)-\frac{n}{\tau \left(T\right)}-\sigma fn+D\left(T\right)\left(\frac{\partial^{2}n}{\partial x^{2}}+\frac{\partial^{2}n}{\partial y^{2}}+\frac{\partial^{2}n}{\partial z^{2}}\right)$$

For efficiency reasons, the calculation of the 3D temperature profile due to beam heating and the associated temperature dependency of surface diffusion and residence time are optional. It may be deactivated, e.g., for deposits with good thermal conductivity. 

Further in this text, a simulation executed with constant diffusion coefficient and residence time is regarded as *temperature untracked*. A simulation executed with calculation of a temperature profile and calculation of temperature-dependent precursor coverage is regarded to as *temperature tracked*.

### 2.5. Parameters

In this work, we focus on the well-established and widely used precursor Me_3_PtCpMe ((CH_3_)_3_Pt-cyclopentadienyl-CH_3_), assuming a deposit composition of PtC_5_. The initial substrate material is assumed to be gold. All of the discussed structures were deposited using 30 keV acceleration voltage and 0.15 nA beam current as typical values for high-resolution growth. The precursor flux *Φ* at the surface was calculated to be 1700 nm^−2^·s^−1^ for our gas injection system: (14)$$\Phi =\frac{P}{\sqrt{2\pi m_{g}k_{b}T_{g}}}$$
where *P* is gas pressure, *m_g_* is the gas molecule mass, *k_B_* is the Boltzmann’s constant and *T_g_* is gas temperature [28]. The equilibrium surface precursor coverage resulting from our simulations without temperature tracking is 0.160 molecules/nm^2^.

The parameters for the simulation are collected in Table 1. Simulations without temperature tracking assume a surface diffusion coefficient of 400,000 nm^2^/s and mean residence time of 160 µs for MeCpPtMe_3_ adsorbed on the PtC_5_ surface. The temperature dependence of the residence time of Me_3_CpPtMe on a SiO_2_ substrate has been studied in [27] and later adopted for adsorption/desorption on PtC_5_ deposits [20], resulting in an activation energy of 0.62 eV. In our simulations, the best correspondence with the experiments was obtained with a slightly lower value of 0.565 eV. The surface diffusion exponential pre-factor and activation energy were taken from the literature [26]. Here again, the activation energy was lowered from 122 meV to 0.098 meV. These parameters were also used in the initial growth stage on the golden substrate due to the quick formation of a thin film of PtC_5_ at process start. The thermal conductivity was raised from 0.16 [20] to 0.8 W/m/K, which resulted in the best reproduction of the temperature profile from the same work [20].

The electron scattering model requires the knowledge of the atomic number in Equations (3) and (4). Compound materials are conventionally characterized by an effective or average *Z_av_*. In this current work, *Z_av_* is determined by calculating the root mean square (RMS) average *Z_RMS_* and atom number density *n_a_* of the constituting elements and then choosing the *atomic number* of an element from the periodic table of elements that has the closest values. This technique has been shown to yield simulated BSE yields close to that of experiments with simple metallic compounds [35].
(15)$$n_{a}=\frac{n_{m}N_{A}\rho }{M}\;\;$$
where *n_m_* is the number of atoms in a molecule, *ρ* is the mass density, and *M* is the molecular mass. For the PtC_5_ deposits, we obtain *n_a_ = 6.5 (10^22^* atoms/cm^3^*)* and *Z_RMS_ = 29.89*. The closest element to these values is Zn with *Z = 30* and *n_a_ = 6.5 (10^22^* atoms/cm^3^*).*

The beam profile is described by a super-Gaussian probability distribution (Figure 2) and allows to roughly approximate beam defocus by varying *σ* and n:(16)$$p(r)=\exp\left(-2\cdot {\left(\frac{r^{2}}{\sigma^{2}}\right)}^{n}\right)$$
where *r* is the radius taken from the beam center and *σ* is the standard deviation.

### 2.6. Experimental

A dual-beam microscope Nova 600 (FEI Company, Eindhoven, The Netherlands) at Goethe University Frankfurt was used for the FEBID experiments. All experiments were performed on Si substrates with a 100 nm Au layer deposited by PVD. The precursor MeCpPtMe_3_ used was preheated to 45 °C for at least 30 min. The working distance during the deposition was 5.2 mm, which included 100 µm distance between the GIS nozzle and the substrate. The GIS nozzle tilt angle was set to 50° relative to the substrate. To prevent any mechanical or beam drift, a waiting time of 10 min was introduced right before the start of the deposition process. Subsequent SEM imaging was performed after a minimum of two hours of pumping time to eliminate all adsorbed precursors and avoid additional deposits. The array of hockey sticks was imaged in a single run at a stage tilt of 52° at highest resolution and a beam current of 21 pA to prevent a change in the opening angle between the pillar and the lateral segment (see below). 

The simulated growth of the five hockey sticks was performed in two steps. Firstly, the patterning files were created and used for the experimental growths. Patterning files were then separated into a stationary (pillar section) and laterally moving section (tilted section). As the vertical pillar segment growth conditions were identical, the simulation of pillar growth section was performed only once and saved as a 3D object file. This file was then used as a start segment for simulating the growth of tilted sections with different patterning velocities. 

### 2.7. Calibration

In search of a 3D structure to be used in our simulation modeling for convergence assessment with the experiment, we looked for a shape that would be simple, controllable and exhibit a prominent response to changing deposition conditions and precursor parameters. The chosen shape consists of a vertical pillar and a tilted segment attached to the top of the pillar. Such a “hockey stick” structure with different configurations has already been utilized in a number of works concerning the study of the FEBID process [4,20,23,36,37,38] as it demonstrates the two key process features: vertical and lateral growth. The geometrical features, such as pillar height, segment thickness, length and profile, tilted segment angle and curvature, as well as co-deposit features, enable a detailed analysis of several aspects of the growth process at once.

In order to test, validate and identify the limitations of the simulation, a series of preliminary experiments was carried out. For the purpose of testing the 3D simulation, the hockey stick structure was used with varying heights, tilted segment angles and lengths. 

In total, five hockey sticks were deposited with lateral patterning velocities of 40, 50, 70, 90 and 110 nm/s to assess the lateral growth capabilities and control over the hockey stick angles. All of them feature the same exposure time for the initial pillar (7s) and length of the tilted segment (500 nm) in horizontal x–y plane. The deposition was carried out in one session with 30 keV beam energy and 0.15 nA beam current. Due to the incomplete dissociation of the precursor molecules, structures deposited with Me_3_PtCpMe tend to lose volume under extended beam exposure. We have observed a decrease in the segment tilt angle by ~3° between consequent scans of the hockey stick with the largest tilt angle (lateral patterning velocity 40 nm/s). Experimentally grown structures and simulation results are presented in Figure 3. Simulations took 4.5, 4, 3.5, 2 and 1 h for the 5 hockey sticks using a 5 nm cell grid with a deposition scaling factor of 4 (see Section 3.1). 

Temperature effects were simulated by depositing a hockey stick with 1.3 µm tilted segment length with temperature tracking on a 10 nm cell grid. The result was compared with the same simulation configuration but with temperature tracking disabled.

**Table 1 nanomaterials-13-00858-t001:** Me_3_PtCpMe precursor and PtC_5_ deposit properties critical to the simulation.

Symbol	Description	Value	Units	Reference
n_0_	Maximum surface precursor density	2.8	1/nm^2^	[6]
dV	Effective dissociated precursor molecule volume	0.094	nm^3^	
σ	Integral precursor cross-section	0.022	nm^2^	[39]
D	Mean surface diffusion coefficient	400,000	nm^2^/s	[26]
τ	Mean residence time	100	µs	[6]
ε	Energy required to create a secondary electron	73	eV	[34]
λ	SE mean free path	2.5	nm	[34]
Z_av_	Average atomic number	30		
M	Average molecular mass	42.5	g/mole	
ρ	Density	4.5	g/cm^3^	[40]
k_0_	Residence time preexponential factor	10^13^	Hz	[27]
E_a_	Adsorption activation energy	0.565	eV	[20]
D_0_	Surface diffusion preexponential factor	4,200,000	nm^2^/s	[26]
E_D_	Surface diffusion activation energy	0.098	eV	[26]
k	Thermal conductivity	0.8	W/m/K	Curr. work

## 3. Results

The hockey sticks with varying segment angles (Figure 3) show acceptable agreement between the simulation and experiment in terms of segment angle and length. Simulated structures exhibit an oval cross-section of the tilted segment, which is more pronounced for smaller tilt angles. The pronounced height increase for larger angles of the hockey sticks grown by the simulation can be explained by the increasing temperature with the increasing length of the tilted segment. The middle (blue) hockey stick is in best agreement with the experimental result in terms of inclination angle, while others show a noticeable deviation. 

The effect of activating temperature tracking on the shape of the tilted segment is depicted in Figure 4. The characteristic down curvature observed in the experimental results is very well reproduced in the simulation and is a consequence of the temperature-dependent surface diffusion and residence time. The shown temperature profile corresponds to the moment when the simulation was completed. The simulation also reproduced the formation of the co-deposit on the substrate just below the tilted segment. By retrieving the beam position from the saved data and performing a static Monte Carlo simulation during the tilted segment growth stage, the co-deposit formation can be observed (see Figure 5). Figure 4a shows that the scattering of the beam in the tilted segment is strong enough to exclude localized co-deposit formation by transmitted electrons. All electron trajectories already exhibit significant scattering at a penetration depth of about 20 nm into the tilted segment. Localized co-deposit growth is a consequence of the decreasing lateral growth speed due to the rising temperature, which causes the beam spot to be ahead of the growth front and, thus, partially hit the substrate. 

Figure 6 represents data collected during the simulation run. As seen from the graphs, the growth rate and precursor coverage evolution exhibit additional features in temperature-tracked mode. The hockey stick grown without temperature tracking is regarded as Structure 1 and, with temperature tracking, as Structure 2. The pillar growth stage is characterized by accelerating growth due to electron saturation. At a certain pillar height, the growth rate reaches its maximum. While the growth rate of Structure 1 remains at a plateau during this stage, Structure 2 experiences a decline due to the rising temperature. During the next transient stage, the beam starts to dislocate laterally and establishes the connecting base of the lateral segment. At this stage, the growth rates of both structures spike as the beam irradiates a significant area on the side of the pillar resulting in higher surface SE yield. The temperature of Structure 2 decreases at this stage as the distribution of heat sources induced by the beam moves temporarily closer to the base of the structure. 

At the last stage, the tilted segment growth continues. For Structure 1, the growth rate and precursor coverage at the BIR quickly arrive at a plateau. A slight drop in precursor coverage can be observed despite the fact that temperature tracking is disabled. Due to the fact that both beam parameters and diffusion coefficient are constant, it is evident that the precursor coverage drop is caused by another changed growth condition which is the surface configuration at the BIR. Note, however, that this effect is not observable during the growth of Structure 2 as it masked by the precursor coverage spike caused by the temperature drop. 

The temperature-untracked mode is characterized by a depleted area around the BIR that exists only in the vicinity of the deposition area. This is clearly seen from the BIR precursor coverage profile in Figure 7. In contradistinction, the temperature profile of Structure 2 exhibits a precursor coverage gradient on the whole surface of the structure as a result of the established temperature gradient. A comparison of the equilibrium precursor coverage at (327 K) to the observed value at the tilted segment tip reveals that depletion in the BIR center is only 20%, which is 3 times less than at the start of the deposition. 

To summarize this part, the bending of the structure has a multi-factorial cause. The beam is heating the sample due to Joule heating. The transfer of the heat to the sample at the growth front and the heat dissipation at the sample base via the substrate induces a steady-state temperature gradient. During the growth, the temperature has an increasing trend as the structure grows longer and increases its thermal resistance. The thermally activated dissociation and diffusion increase their rates in response to the rising temperature, directly affecting precursor coverage and growth rate. The interplay of decreasing residence time and increasing surface diffusion coefficient governs the growth dynamics and eventually causes the curvature (see Appendix A). 

### 3.1. Performance

Simulation performance is one of the features that defines not only the ease of use of a digital tool but also the tasks that can be accomplished in a reasonable amount of time. Thus, several optimizations and settings were introduced to the simulation for time efficiency improvement. Programming solutions include the ray tracing algorithm for electron scattering (see Appendix A), vectorized computations and translations of the most computationally intensive blocks to C. Further optimizations could include parallelization of the Monte Carlo module as each electron trajectory is treated independently. Finally, the computing power of GPUs could be utilized. 

A technique used in this work is an acceleration of the virtual growth. During a simulation run, at each time step, the deposited volume for each irradiated cell is calculated and then added to the total deposited volume of the structure. By introducing a scaling factor *g* > 1 by which each deposited volume at each time step is multiplied, the deposited volume is increased, and thus deposition is accelerated. Consequently, the time required to deposit a given volume is reduced. As a consequence, each dwell time has to be divided by the scaling factor accordingly. This approach leads to a reduction in the total time steps needed to deposit the same structure at the expense of numerical accuracy regarding solving the differential equation. However, it is evident from Figure 8 that hockey sticks deposited with significantly differing scaling factors do not vary in shape, and the volume difference is 0.1% which is well within the volume variance margin (*Accuracy* section) while significantly reducing the simulation time (Figure 9).

Another opportunity for performance optimization is a tradeoff between simulation speed and grid resolution. Selecting between different cell sizes allows setting coarser or finer grids which affect simulation speed significantly. Figure 9 shows 2 orders of magnitude faster results between 2 and 10 nm cell sizes.

Already in the present state of the simulation, the performance with a sufficiently fine grid size of 5 nm opens a path to various usages of the simulation, including pattern testing and precursor parameter estimation and tuning. Moreover, 3D-object files generated by the simulation can be inspected in detail for precise shape and volume determination of the outcoming nanostructure. They can also be used as a starting structure for further simulations. A series of such files containing precursor coverage data, compiled in series, depicts structure growth live and reveals in situ details of the FEBID process, such as co-deposit formation.

### 3.2. Statisctical Accuracy

For the quantification of the resulting effective accuracy, we used the variance of the total deposited volume during simulation runs with the same parameters and pattern file. In particular, the simulated growth of each of the five hockey sticks was repeated five times in order to evaluate the volume variance. We observed that the variance rises linearly with the deposited volume, i.e., it keeps a constant 0.5% ratio. The deposition scaling factor *g* increases the volume variance primarily due to the decreased number of Monte Carlo draws during a simulation. The MC module is invoked each time the loop detects a newly filled cell, and thus the number of Monte Carlo draws is limited to the number of cells. The growth rate may be significant enough to cause two cells to be filled at one time step. In this case, the MC module runs only once for two filled cells, slightly reducing accuracy. Accelerated growth increases these chances and the number of cells to be filled at the same time. This effectively reduces the number of MC draws per simulation and consequently reduces accuracy.

## 4. Conclusions

FEBID has proved itself as a versatile nano-fabrication capable of yielding structures from simple 2D shapes to complex 3D. In this work, an effective 3D Monte Carlo-continuum simulation is presented, which allows us to predict the actual shape of 3D structures with thermal effects fabricated by FEBID using a given pattern file.

A ‘ghost cell’ method was applied for the numerical treating of surface diffusion, which is a suitable and simple approach to define complex Dirichlet boundary conditions and can be easily combined with the finite difference approach used here. 

Beam heating through primary and secondary electrons is introduced to the continuum model, which allows the tracking of the structure’s 3D temperature profile. The approach assumes immediate relaxation to the steady-state temperature profile inside the growing structure, as the relaxation time is much faster than a typical dwell time. Accordingly, the temperature profile is derived via a numerical relaxation method (SOR). The simulated growth with and without temperature-dependent properties clearly shows a significant impact of beam heating on the angled segment of the hockey stick, which was used as a reference structure. The data collected from simulations with and without temperature tracking reveal the influence of the surface temperature on the precursor coverage across the whole surface. When compared with experiments, the simulation predicts the shape, volume and segment angle of the hockey sticks very well and is also able to reproduce the formation of co-deposit. 

Our simulation provides a versatile tool for 3D structure growth prediction, growth parameters retrieval, informative visual representation of the grown structures and a rigorous control over the simulation setup. Included features make the 3D multi-length scale more tractable and give an opportunity to inspect the complex interplay of electron beam–matter interaction, continuously changing precursor coverage and deposition process.

## Figures and Tables

**Figure 1 nanomaterials-13-00858-f001:**
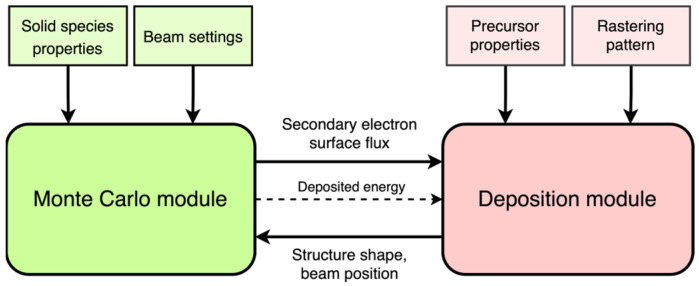
Schematic representation of the simulation design.

**Figure 2 nanomaterials-13-00858-f002:**
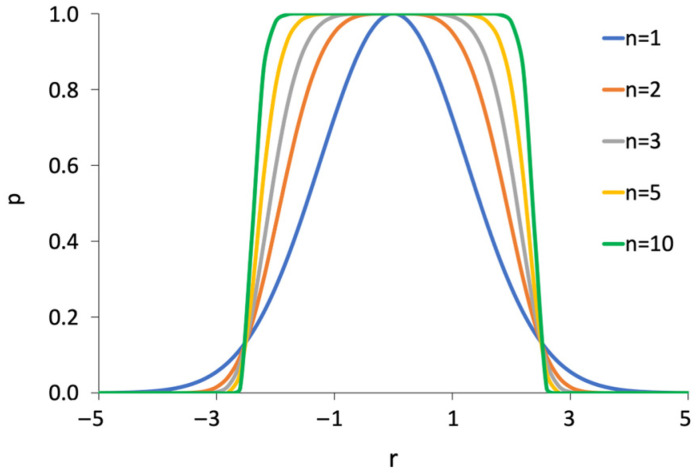
Super-Gaussian distribution.

**Figure 3 nanomaterials-13-00858-f003:**
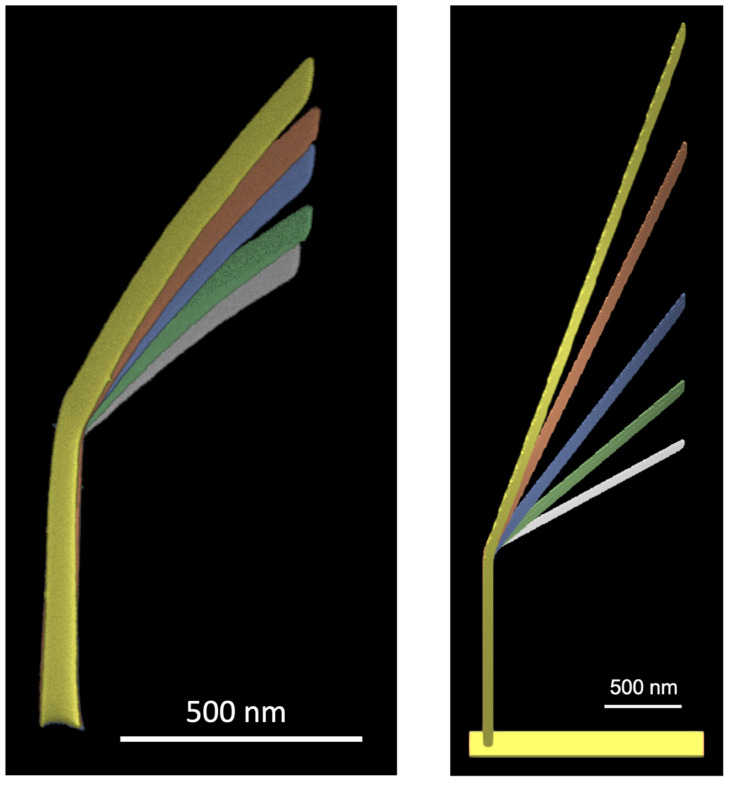
Colored SEM viewgraph of deposited hockey stick array (**left**) and result of the corresponding simulation using the same pattern files (**right**). Simulations were carried out without temperature tracking. Coloring codes and the patterning velocity: white—110 nm/s, green—90 nm/s, blue—70 nm/s, red—50 nm/s, yellow—40 nm/s.

**Figure 4 nanomaterials-13-00858-f004:**
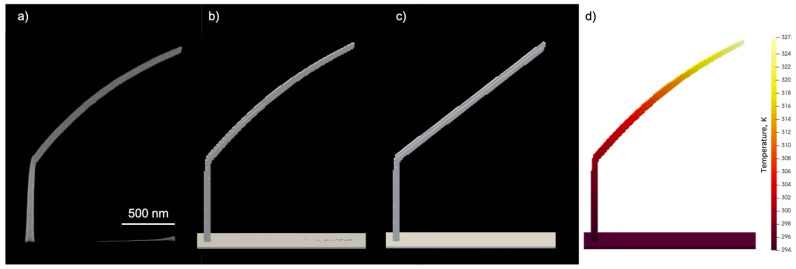
Hockey sticks deposited (**a**) experimentally, (**b**) via simulation with temperature tracking activated, (**c**) via simulation without temperature tracking. (**d**) Temperature profile of the simulated hockey stick recorded at the end of the simulation. Pictures a, b and c correspond to a 52° tilt.

**Figure 5 nanomaterials-13-00858-f005:**
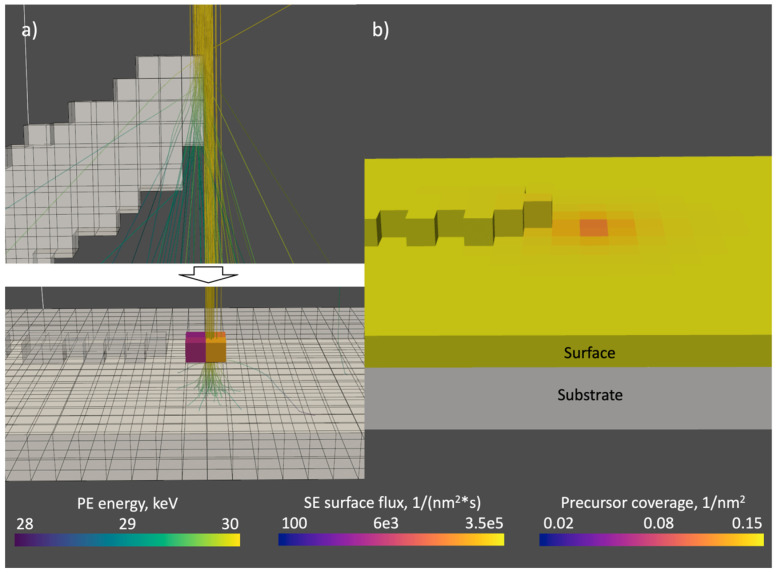
Illustration of co-deposit formation: (**a**) result of static Monte Carlo simulation of the electron beam and resulting SE flux on the substrate surface (grey cells are solid), (**b**) precursor coverage profile of the same area. Cell size 10 nm. The main contribution to co-deposit formation is due to primary electrons that do not enter the growing structure at the growth front. Precursor depletion indicates the growth in front of the co-deposit.

**Figure 6 nanomaterials-13-00858-f006:**
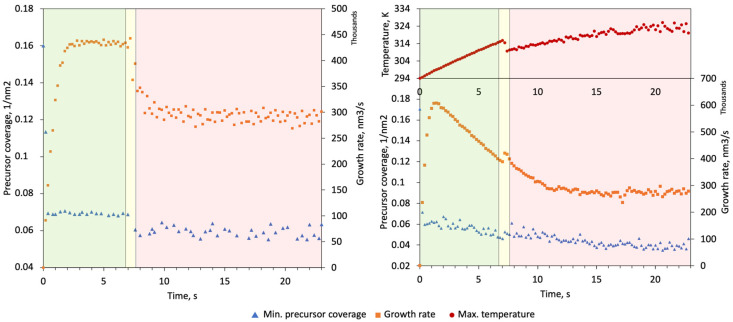
Change of precursor coverage and growth rate during two simulated growths of the same hockey stick. Data on the left was collected with temperature tracking disabled. Data on the right was obtained with temperature tracking enabled during the simulation. The precursor coverage value plotted refers to the respective smallest value on the whole surface and, ultimately, corresponds to the center of the BIR. The growth rate is calculated as the total volume deposited during a time step. The increasing scatter of the recorded temperatures is due to the decreasing number of heat source cells as the tip gets thinner, which is effectively amplified by the rising thermal resistance of the structure.

**Figure 7 nanomaterials-13-00858-f007:**
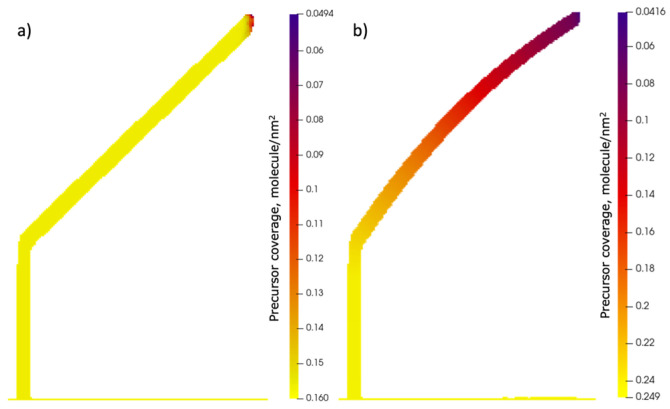
Surface precursor coverage profiles for simulated growth (**a**) without temperature tracking and (**b**) with temperature tracking. Profiles are recorded at the end of the simulation.

**Figure 8 nanomaterials-13-00858-f008:**
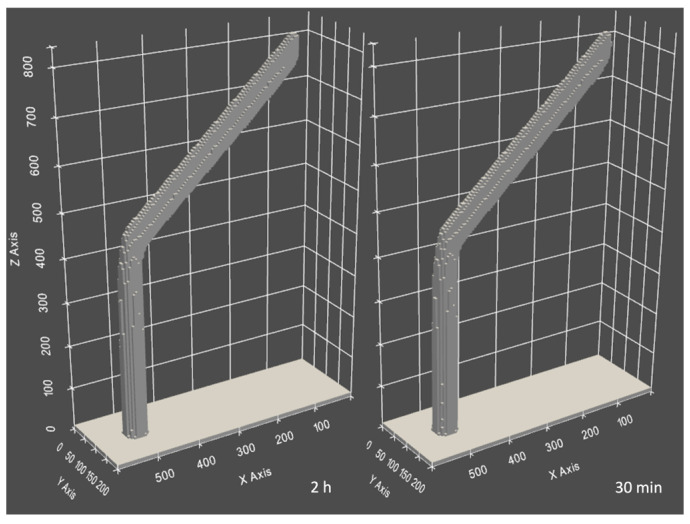
Comparison of hockey sticks deposited with 100 nm/s patterning speed with *g* = 1 (**left**) and *g* = 16 (**right**)*. The lower left label on the picture is the total run time of the simulation. *PC—Intel 8-core i7-7700HQ 2.8GHz, 32Gb RAM.

**Figure 9 nanomaterials-13-00858-f009:**
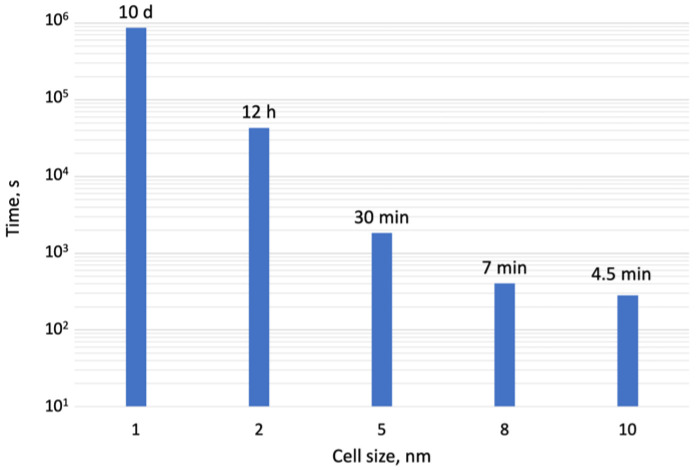
Total time of simulated growth of a 500 nm pillar with different grid discretization on a PC*. *PC—Intel 8-core i7-7700HQ 2.8GHz, 32Gb RAM.

## Data Availability

The data presented in this study are available on request from the corresponding author.

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
