# Peer review of "Fast and Efficient Simulation of the FEBID Process with Thermal Effects"

_nanomaterials, 2023, doi:10.3390/nano13050858_

Round 1
Author Response
We would like to thank the reviewer for his/her overall very positive reception of our work and would like to address the raised minor criticism.
- Abbreviate forms (lowercase keys) mentioned in keywords will mislead the readers, need to revise them.
Our reply: This has been corrected. - Several mistakes were found, so the whole manuscript required a thorough check by the authors to correct typo errors to improve manuscript quality.
Our reply: The manuscript has been thoroughly checked for typos and also broken references. - In section 2.4 Beam heating context, generated/mentioned as “Error! Reference source not found”. Check and revised accordingly.
Our reply: see our reply to revision request 2. - Figures are not clear, figures quality should be improved.
Our reply: We have improved the figure quality and will make sure that it is sufficient for reproduction in the published article.
Reviewer 2 Report
The authors Kuprava et al, reported Fast and efficient simulation of the FEBID process with thermal effects. Although, this work contains some results, the organization and interpretation of the result should be enhanced further. Hence, I recommend this work required a substantial revision before considering for publications.
1. Provide the obtained results in the abstract. The current form seems to be like broad.
2. Several typo errors and English language should be checked before res-submitting to other suitable journal.
3. Novelty of the work should be highlighted in the introduction in more clearly.
4. Author should include more results and data in the abstract.
5. Quality of images are poorly presented. It should be nicely presented. In Figure 2, no scale bar itself.
6. The authors designed the work nicely, merely presented the results but failed to discuss the observed results elaborately.
7. I suggest the authors to compare the previous literature similar to that work to find a merits of this work.
8. Some statement should be revised with proper citation; Applied Catalysis B: Environmental 316 (2022) 121603; Chemical Engineering Journal 446 (2022) 137045.
Reviewer 3 Report
I think the article can be published after minor revision:
MINOR:
Introduction Section
- Introduction section is good written and precisely focused on Monte-Carlo simulations of FEBID. In my opinion, Introduction section needs a couple of paragraphs outlining the importance of hockey stick modelling. Is it a benchmark test for FEBID? Why other structures haven’t been considered.
- Same comment regarding selection of MePtCpMe3 precursor compound and deposition product PtC5. Please indicate a full chemical formula of the precursor. What was your motivation to fabricate a hockey stick of PtC5?
Results
- It’s unclear from the text how the final composition of experimentally grown hockey stick was confirmed. Whether was X-Ray Diffraction conducted?
- Fig. 5 shows a hockey stick grown experimentally and simulated. They look almost same. If I go back to Fig. 3 then I clearly see a significant discrepancy. The experimentally grown stick is twice shorter than one simulated. Why the numerical simulation failed to simulate sizes of hockey sticks in Fig.3? Even without temperature tracking the sizes seem to be fine in Fig. 5.
TYPOS:
- Page 6, line 265: «Error! Reference source not found.. », this note needs to be removed
- Page 11, line 445: missing space, «327 K»
- Page 13, line 490: information under * should be incorporated in the text or footnote
